# New Frontiers of Extracorporeal Shock Wave Medicine in Urology from Bench to Clinical Studies

**DOI:** 10.3390/biomedicines10030675

**Published:** 2022-03-15

**Authors:** Po-Yen Chen, Jai-Hong Cheng, Zong-Sheng Wu, Yao-Chi Chuang

**Affiliations:** 1Department of Urology, Kaohsiung Chang Gung Memorial Hospital, Chang Gung University College of Medicine, Kaohsiung 833, Taiwan; patrick7613@gmail.com (P.-Y.C.); a8905114@gmail.com (Z.-S.W.); 2Center for Shock Wave Medicine and Tissue Engineering, Kaohsiung Chang Gung Memorial Hospital, Chang Gung University College of Medicine, Kaohsiung 833, Taiwan; cjh1106@cgmh.org.tw; 3Graduate Institute of Human Sexuality, Shu-Te University, Kaohsiung 833, Taiwan; 4Division of Medical Research, Kaohsiung Chang Gung Memorial Hospital, Chang Gung University College of Medicine, Kaohsiung 833, Taiwan; 5Department of Leisure and Sports Management, Cheng Shiu University, Kaohsiung 833, Taiwan

**Keywords:** shock waves, cryoinjury, bladder, erectile dysfunction

## Abstract

A shock wave (SW), which carries energy and propagates through a medium, is a type of continuous transmitted sonic wave that can achieve rapid energy transformations. SWs have been applied for many fields of medical science in various treatment settings. In urology, high-energy extracorporeal SWs have been used to disintegrate urolithiasis for 30 years. However, at lower energy levels, SWs enhance the expression of vascular endothelial growth factor (VEGF), endothelial nitric oxide synthase (eNOS), proliferating cell nuclear antigen (PCNA), chemoattractant factors, and the recruitment of progenitor cells, and inhibit inflammatory molecules. Low energy extracorporeal shock wave (LESW) therapy has been used in urology for treating chronic prostatitis/chronic pelvic pain syndrome (CP/CPPS), interstitial cystitis/bladder pain syndrome (IC/BPS), overactive bladder, stress urinary incontinence, and erectile dysfunction through the mechanisms of anti-inflammation, neovascularization, and tissue regeneration. Additionally, LESW have been proven to temporarily increase tissue permeability and facilitate intravesical botulinum toxin delivery for treating overactive bladders in animal studies and in a human clinical trial. LESW assisted drug delivery was also suggested to have a synergistic effect in combination with cisplatin to improve the anti-cancer effect for treating urothelial cancer in an in vitro and in vivo study. LESW assisted drug delivery in uro-oncology is an interesting suggestion, but no comprehensive clinical trials have been conducted as of yet. Taken together, LESW is a promising method for the treatment of various diseases in urology. However, further investigation with a large scale of clinical studies is necessary to confirm the real role of LESW in clinical use. This article provides information on the basics of SW physics, mechanisms of action on biological systems, and new frontiers of SW medicine in urology.

## 1. Introduction

Extracorporeal shock wave lithotripsy (ESWL) was the first application of SW medicine in humans [1]. Weinstein et al. demonstrated that SWs have loosening effects on the bone-cement interface for revision arthroplasty in dogs [2]. Thereafter, ESWT has been widely used and studied in musculoskeletal disorders [3,4]. Additionally, cardiac SW therapy has anti-anginal effects, increases exercise capacity, and improves myocardial perfusion and clinical symptoms in the treatment of patients with stable coronary artery disease with a safety profile [5]. ESWT was also shown to enhance skin flap survival through increasing blood perfusion and neovascularization in a compromised skin flap model in rats [6]. Thereafter, ESWT was suggested as an effective and safe treatment for soft tissue wounds [7,8], and was approved for treating diabetic foot ulcers by the FDA in 2017 [9].

ESWT has been widely studied in various new fields of biomedicine [10,11]. New findings and indications of ESWT for urology have increased in recent years [12,13,14]. This paper reviews mechanisms of action and clinical results of ESWT on new frontiers in urology.

## 2. Characteristics of SWs

The SW is an acoustic wave, similar to the sound of thunder in nature, or the sound of a supersonic aircraft. A SW can be used to transfer energy into the human body to restore body function. The characteristics of SWs are different from those of an ultrasound wave (Figure 1A). The SW is a biphasic supersonic wave that has high pressure amplitude, a thousand times more than that of an ultrasound wave [15]. When a SW is generated, a high pressure wave is produced with a value of approximately 100 megapascals (MPa) in 10 nanoseconds, followed by a negative pressure value of approximately negative 10 MPa in microseconds (Figure 1B) [15]. Three types of generators can be used to produce SWs, including electrohydraulic, electromagnetic, and piezoelectric. Radial extracorporeal shock wave (rESW), a ballistic pressure wave (approximately with 1 MPa in microseconds), is produced by a compressed air device. The characteristics of rESWs are different from those of acoustic SWs (Figure 1C). There are four forms of SW, including focused, defocused, planar, and radial, which are generated depending on the source and the design of the devices. SWs carry energy through the human body and induce physical, chemical, and biological effects [16]. Focused, defocused, and planar SW treatments are called ESWT. The energy of LESWs used for tissue regeneration is approximately 5 Mpa, which is lower than that of ESWL (above 100 MPa) used to disintegrate kidney stones.

## 3. Dosage of ESWT

The energy flux density (EFD) is used as the parameter of ESWT to measure the energy per square area (mJ/mm^2^) that transmits the energy of the SW into tissue [15]. There is currently no consensus on the boundary values of high, medium, and low energy SWs. Different manufacturers or studies define the range of the energy into energy bands. Indeed, in 1998, high (>0.6 mJ/mm^2^), medium (0.08–0.28 mJ/mm^2^), and low energy (>0.08 mJ/mm^2^) of ESWT was defined by Rompe at al. to define dose-related effects of SWs on rabbit tendo Achillis [17]. A review article on ESWT for the treatment of soft tissue conditions considered low energy ESWT as EFD ≤ 0.12 mJ/mm^2^, and high energy as >0.12 mJ/mm^2^ [18]. Bannuru et al. suggested that high energy (≥0.28 mJ/mm^2^) SWs better relieve pain and improve function in chronic calcific shoulder tendinitis compared to low energy (<0.28 mJ/mm^2^) SW therapy [18]. Wang and Lue et al. defined low intensity extracorporeal SW therapy (Li ESWT) as a form of energy transfer that is <0.2 mJ/mm^2^ than the ESWT employed for lithotripsy and nephrolithiasis treatment [19]. Therefore, there remains no agreement to the definition of high, medium, and low energy for ESWT.

## 4. Biological Effects of ESWT

The biological effects and related energy levels of ESWT are presented in Figure 2. Haupt proposed that the mechanisms of action of ESWT have four reaction phases [20], including physical, physical-chemical, chemical, and biological phases. The generation of SWs induces a positive pressure wave, followed by a negative pressure wave to pass through the medium. A portion of the wave undergoes absorption, reflection, and refraction [15]. Additionally, a negative pressure wave produces a tensile force and induces cavitation and ionization in the liquid after a positive pressure wave. These physical forces increase cell membrane permeability and release biomolecules, such as ATP or ions, in the physical-chemical and chemical phases [21,22,23]. Finally, mechanotransduction after SW treatment induces the release of exosomes, microRNA, growth factors, cytokines, chemokines, and other molecules from different cells to produce various biological effects [16,24,25,26].

Many biological effects and related molecules of ESWT were reported, including anti-inflammation, immunomodulation, angiogenesis, cell proliferation and differentiation, stem cell attraction, exosome release, nerve regeneration, and increase in cell membrane permeability (Figure 3) [13,16,25,27,28,29,30].

## 5. Anti-Inflammation

ESWT reduces expression of tumor necrosis factor alpha (TNF-α), COX2, nuclear factor kappa B (NF-κB), and chemokines (CC and CXC), all of which contribute to anti-inflammation reactions [31]. ESWT induces expression of G-protein coupled receptor 120 (GPR120), which has been shown to inhibit transforming growth factor beta-activated kinase 1 (TAK1) and NF-κB signaling to modulate the inflammatory response in a cystitis model in rats [32]. Sukubo et al. demonstrated that macrophage exposure to low energy SW dampens the induction of the pro-inflammatory profile characterizing M1 macrophages and promotes the acquisition of an anti-inflammatory profile synergizing with macrophage alternative activation, which is associated with a significant increase in IL-10 and a reduction in IL-1β production [33]. Holfeld et al. revealed that ESWT modulates inflammation via the TLR3 pathway, including the interaction between interleukin (IL)-6 and IL-10 in TLR3 stimulation as a three-phase regulation over time in a study of ESWT application to human umbilical vein endothelial cells (HUVECs) [34]. ESWT also has effects on the suppression of pro-inflammatory cytokines, chemokines, and matrix metalloproteases (MMPs), as well as the attenuation of leukocyte infiltration for treating wound healing, severe cutaneous burn injury, and cystitis models in animals [35,36,37,38].

## 6. Angiogenesis

Wang et al. investigated the effect of ESWT on neovascularization at the tendon–bone junction in a rabbit model [24]. The results demonstrated that ESWT produces a significantly higher number of neo-vessels and angiogenesis-related markers, including eNOS, VEGF, and PCNA, compared to the control. Another study showed that treatment with multiple sessions of ESWT significantly enhanced diabetic wound healing, which was associated with increased neovascularization and tissue regeneration, correlating with VEGF and the mitogen-activated protein kinase-mediated pathway [39]. It was also demonstrated that ESWT enhanced the expression of various angiogenesis-related factors, including VEGF, IL8, stromal cell-derived factor 1 (SDF-1), eNOS, CXC motif chemokine 4 (CXCR4), and basic fibroblast growth factor (bFGF), improving tissue perfusion in clinical trials and animal models [40,41,42]. Huang et al. suggested that ESWT enhanced angiogenesis via ligand-independent activation of VEGFR2, and further prolonged angiogenesis through endosome-to-plasma membrane recycling in HUVECs [43].

## 7. Cell Proliferation

Cell proliferation is a major necessity for tissue regeneration. ESWT induces up-regulation of VEGF and IL-8, promotes the proliferation of endothelial progenitor cells, and improves myocardial function in patients with refractory coronary artery disease [42]. ESWT was proven to trigger the release of cellular ATP, which subsequently activates purinergic receptors and finally enhances proliferation in vitro and in vivo via downstream of Erk1/2 signaling [22]. Additionally, ESWT activates the mTOR-FAK mechanotransduction signaling axis and enhances the proliferation of mesenchymal stem cells (MSCs) [44].

## 8. Cell Membrane Permeability

ESWT, inducing a shear force and cavitation through the liquid on cells, can temporarily increase the cell membrane and tissue permeability, and facilitate drug delivery or gene therapy [45,46]. Kung et al. reported that ESWT transiently opens the blood-brain barrier and delivers the genes into brain with no detrimental effects [47]. Our previous study demonstrated that ESWT induces significant down regulation of E-cadherin and ZO-1, increases the permeability of the bladder urothelium, and facilitates intravesical botulinum toxin delivery in a rat model [48]. Moreover, Ito et al. evaluated the interferometric and fluorescence analysis of SW effects on the cell membrane, the results of which suggested a possible mechanism of membrane permeation where sharp pressure gradients create pores on the membrane, as evidenced by the transport of fluorescent dye molecules from outside to inside the cells following shock-induced membrane permeability [30]. Additionally, LESW-assisted drug delivery was suggested to have a synergistic effect in combination with cisplatin, improving the anti-cancer effect by improving chemo-agent tissue infiltration and decreasing cisplatin efflux membranous protein expression for treating urothelial cancer in an in vitro and in vivo study [49].

## 9. Nerve Regeneration

Hausner et al. demonstrated that ESWT improves the rate of axonal regeneration in a sciatic nerve injury model in rats, likely via faster Wallerian degeneration and improved removal of degenerated axons and regenerated injured axons with greater capacity [27]. Lee et al. demonstrated that ESWT significantly increased the sciatic functional index score, reduced the level of muscle atrophy, reordered the injured nerves, and activated the conjunction of the muscle and neurons in a sciatic nerve-crushing damage model in rats [50]. The rESW enhances neural stem cell proliferation and differentiation by activation of the PI3K/AKT, Wnt/β-catenin, and Notch pathway [51]. Sensing of tissue damage and subsequent orchestration of the inflammatory response via TLR3 were suggested to represent an innate mechanism of tissue regeneration, which was proven to be involved in neuroprotection and spinal cord repair in a mice spinal cord contusion model treated with ESWT [52]. ESWT was also shown to increase the expression of brain-derived neurotrophic factor (BDNF) and activate the PERK/ATF4 signaling pathway to induce nerve regeneration after nerve injury [53]. However, the detailed mechanism and more evidence-based results and clinical studies should be investigated in the future.

## 10. Application of Low Energy Shock Wave (LESW) in Urology

### 10.1. Chronic Prostatitis/Chronic Pelvic Pain Syndrome

Chronic prostatitis/chronic pelvic pain syndrome (CP/CPPS) is an inflammatory disease, with an estimated prevalence from 8.4% to 25% in different studies [54,55]. According to the National Institutes of Health (NIH) categories of prostatitis, type III chronic abacterial prostatitis, including type IIIa–inflammatory CP/CPPS and type IIIb—noninflammatory CP/CPPS, and type IV—asymptomatic inflammatory prostatitis, meet the definition of CP/CPPS. It was suggested that the pathophysiology of CP/CPPS involves nerve growth factor (NGF) and IL-10, which directly correlated with pain severity and the inflammatory process [56,57]. As no single modality was proven to be effective with long-term durability for treatment of CP/CPPS, ESWT was investigated as a promising alternative in recent decades. All the studies are summarized in the Table 1.

Jeon et al. reported that ESWT improved CP/CPPS and reduced inflammation by degrading COX-2 in the microenvironment through the TLR4-NFκB-inhibiting pathway in lipopolysaccharide-induced inflammation in RWPE-2 cells and a 17 β-estradiol and dihydrotestosterone-induced prostatitis rat model [74].

Wang et al. demonstrated that LESW significantly suppressed the expression of IL-1β, COX-2, caspase-1, and NGF on day 3, and IL-1β, TNF-α, COX-2, NALP1, caspase-1, and NGF expression on day 7, in a dose-dependent fashion in a capsaicin-induced prostatitis model in rats [75]. A recent study revealed that ESWT decreased the number of total and degranulated mast cells and alleviated pelvic pain in a rat model of prostatitis induced by intraprostatic injection of 1% carrageenan [76]. Taken together, these findings suggest that ESWT can inhibit prostate inflammation and prostatic pain through different mechanisms of action, according to various CP/CPPS animal models.

Zimmermann et al. conducted the first randomized double-blind, placebo-controlled study using ESWT on chronic nonbacterial prostatitis [59]. Thirty patients in the study group received 3000 impulses at 0.25 mJouls/mm^2^, 3 Hz once per week for 4 weeks. The study group demonstrated significant improvement in pain, quality of life (QoL), and voiding function, compared to the placebo group. The various treatment settings of ESWT for treating CP/CPPS are described in Table 1. Taken together, ESWT is an efficient and safe method to decrease the NIH-CPSI score, with therapeutic effects sustained from 12 weeks to 12 months [60,61,65,68,69]. In a recent study including 155 patients, the results showed that 82.8% patients had a ≥6-point decrease in the NIH-Chronic Prostatitis Symptom Index (NIH-CPSI) total score at 1 month; and other functional scores, such as IPSS, VAS, and IIEF-5, were also improved [72]. Wu et al. reported the largest cohort study with long-term follow up [73], which demonstrated that, apart from pain alleviation, ESWT ameliorated the severity of other prostatitis symptoms in a CP/CPPS cohort, with a 53.6% decrease in NIH-CPSI, 17.9% increase in IIEF-5, 6.8% increase in EHS, and 50.9% decrease in IPSS 12 months after ESWT. Although these studies provided clinical evidence for the efficacy of ESWT, the mechanisms of its action in human prostate specimens and biomarkers remained to be determined.

### 10.2. Interstitial Cystitis/Bladder Pain Syndrome (IC/BPS), Ketamine Cystitis, and Radiation Cystitis

IC/BPS is defined as “an unpleasant sensation (pain, pressure, or discomfort) perceived to be related to the urinary bladder, associated with lower urinary tract symptoms of more than six weeks duration, in the absence of infection or other identifiable causes” [77]. Many treatments have been used for IC/BPS based on the different pathophysiology; however, there is currently no established optimal treatment that is considered to be effective long-term without side effects.

In an animal study, LESW showed anti-inflammatory effects in the down regulation of IL-6, COX-2, and NGF expression, and suppressed the bladder overactivity and pain behavior in cyclophosphamide (CYP)-induced cystitis in rats [78]. Chen et al. demonstrated that LESW eased inflammation and oxidative stress by decreasing IL-12, MMP9, TNF-a, nuclear factor-kB, NADPH oxidase 1 (NOX-1) and NOX-2, and iNOS expression in another cyclophosphamide (CYP)-induced cystitis model in rats [36].

Chuang et al. demonstrated the first randomized double-blind, placebo-controlled study using ESWT for 54 patients with IC/BPS. The treatment groups received 2000 shocks, with an energy level up to 0.25 mJ/mm^2^ and frequency of 3 Hz once a week for 4 weeks at the suprapubic bladder area. The patients who received ESWT showed improvements in pain scale and O’Leary-Sant symptom (OSS) scores from baseline to 4 weeks, but only the VAS score reached statistical significance, compared to the placebo group [79]. Shen et al. analyzed 25 eligible patients with IC/BPS from a single center and found that the ESWT group exhibited a significant reduction in the OSS and VAS, compared to the placebo group 4 weeks post-treatment (*p* < 0.05), and the effects were persistent at 12 weeks. The change in the difference of urinary markers in ESWT versus placebo was *p* = 0.054 for IL-4, *p* = 0.013 for VEGF, and *p* = 0.039 for IL-9 at 4 weeks [80].

Chen et al. illustrated a radiotherapy-induced chronic cystitis rat model that was treated with 0.1 mJ/mm^2^/120 impulses every 3 days for a total of 20 times. By day 60, the detrusor contraction was significantly better for the rats receiving LESW. The protein expression of oxidative stress markers, apoptosis markers, DNA-damage markers, inflammatory signaling markers, and fibrosis markers were significantly reduced after LESW treatment [81,82]. The researchers further conducted a ketamine-induced cystitis model treated with 0.12 or 0.16 mJ/mm^2^/120 impulses/at 3 h and 3, 7, 14, 21, and 28 days after ketamine administration. The study demonstrated that LESW effectively attenuated ketamine-induced bladder damage in both molecular and histopathological findings [83]. However, the beneficial effects of ESWT on pre-clinical studies of radiation cystitis and ketamine cystitis warrant that larger multi-institutional studies with placebo control are needed.

### 10.3. Overactive Bladder (OAB)

OAB is a symptom characterized by “urgency with or without urge incontinence, usually with frequency and nocturia” in ICS terminology [82]. Therapies for OAB include behavioral therapies/lifestyle modifications, monotherapy, combinations of antimuscarinics and β3-adrenoceptor agonist, and intradetrusor injection of onaBoNTA, or neuromodulation [84]. However, some patients remain refractory to the above therapies.

LESWs have been studied for treatment of OAB. Lee et al. conducted the first clinical cohort of 82 patients with OAB who received 3000 pulses, 0.25 mJ/mm^2^ and 3 pulses/s per week over an 8-week treatment period. The subjects showed an improvement in OAB symptoms, QoL, and urodynamic parameters during the 3-month follow-up period [85]. Lu et al. reported that the therapeutic effects on OAB symptoms persisted for as long as 6 months. However, further investigation is warranted to confirm the efficacy of ESWT for treating OAB [86].

### 10.4. Erectile Dysfunction (ED)

The pathophysiology of ED is considered multifaceted, with complex mechanisms, including neural, vascular, and hormonal signaling problems. Many chronic diseases, such as hypertension, diabetes, hyperlipidemia, and metabolic syndrome, have been suggested to be associated with ED [87]. Therefore, many studies of ED are important to have solutions to improve the ED after ESWT (Table 2). In the era of phosphodiesterase type 5 inhibitor (PDE5i), oral medication has become the first-line treatment for ED. However, there remain approximately 30–35 % PDE5i non-responders [88,89].

The fundamental mechanisms of LESW for treating ED are still incompletely understood. LESW activates focal adhesion kinase (FAK), extracellular-signal-regulated kinase (ERK), and Wnt signaling pathways, induces cell proliferation, endothelial and smooth muscle restoration, and increases VEGF and eNOS expression [88]. It has been suggested that LESW regenerates penile smooth muscle via the PERK pathway and ATP/P2X7 pathway. Moreover, LESW mediates BDNF activation in penis tissue to sustain neurotrophic healing and induces MSCs to express VEGF.

Muller et al., in a rat model using three sessions of 2000 SWs at 2 BAR and 1000 SWs at 1 BAR on erectile tissue, revealed increased apoptosis and corporal smooth muscle collagenization [111]. On the contrary, the other studies revealed tissue restoration, neuronal nitric oxide synthase (nNOS)-positive nerve generation, and angiogenesis of penile tissue in the diabetes mellitus-associated ED rat model [112,113].

Vardi et al. conducted the first clinical randomized-controlled trial delivering SWs with 300 shocks at 0.09 mJ/mm^2^ on the distal, middle, and proximal penile shaft, and bilateral crura. After a 9-week treatment course, the treatment group revealed significant improvement on the International Index of Erectile Function (IIEF; 6.7 points increase) score, with no significant side effect [94]. Many cohort studies and double-blind, sham controlled studies have been published, which revealed encouraging results. We have summarized the clinical studies in Table 2. However, LESWs have no established definitive recommendations from the EAU guideline for ED treatment due to lack of longer-term follow-up data and unequivocal evidence.

### 10.5. Stress Urinary Incontinence

Stress urinary incontinence (SUI) is currently defined as involuntary loss of urine on abdominal effort, physical exertion, sneezing, or coughing. Treatment modalities include conservative measurement, pelvic floor muscle training, biofeedback, pharmacological, and surgery therapy [114].

Wu et al. described a rat model using vaginal balloon dilation-induced stress urinary incontinence, followed by treatment with 0.06 mJ/mm^2^/300 shocks/3 Hz of SWs. The results showed that LESW eased SUI by promoting angiogenesis, progenitor cell recruitment, and urethral sphincter regeneration [115]. Zhang et al. developed a delayed treatment with LESW in an irreversible rat model of SUI. They found that LESW appeared to increase the smooth muscle content in the urethra and vagina, increase the thickness of the urethral wall, improve striated muscle content and neuromuscular junctions, restore the integrity of the urothelium, increase the number of EdU-retaining progenitor cells in the urethral wall, and improve the continent function [116].

A multicenter, single-blind, randomized-controlled trial study was developed from 60 female patients with SUI who were randomly assigned to receive LESW with 0.25 mJ/mm^2^ intensity, 3000 pulses, and 3 pulses/s, once weekly for a 4-week (W4) or 8-week (W8) period, or an identical sham LESW treatment without energy transmission. They reported that 8 weeks of LESW attenuated SUI symptoms upon physical activity, induced urine leakage, and alleviated OAB symptoms, which implied that LESWT significantly improved QoL [117]. However, a larger scale of the clinical study is necessary to elucidate the real role of LESW for the treatment of SUI.

### 10.6. Detrusor Underactivity/Underactive Bladder

Detrusor underactivity (DU) represents a contraction of reduced strength and/or duration during voiding, resulting in prolonged bladder emptying and/or failure to empty the bladder within a normal timespan [118,119]. The estimated prevalence of DU is 9–28% of men < 50 years, 48% in older men, and 12–45% in elderly women [120]. The pathophysiology of DU is not well understood, but chronic ischemia and atherosclerosis of the microcirculation were mentioned in the literature [121,122]. There is currently no effective treatment for DU.

LESWs have been proven to induce cell proliferation, angiogenesis, and facilitate tissue regeneration. Chuang et al. investigated the effects of LESW in a DU model induced by cryoinjury in female Sprague Dawley rats. The results showed that LESW downregulated IL-6 and COX-2 expression, upregulated α-SMA, induced cell proliferation (evidenced by increasing CD31 and Ki67), increased the contraction amplitude, and improved voiding function [123], suggesting the potential for a non-invasive modality for myogenic DU. Wang et al. treated streptozotocin-induced diabetic rats with LESW 0.02 mJ/mm^2^ at 3 Hz for 400 pulses once per week [124]. The study revealed improved detrusor contractility and reduced post-void residual urine, in association with increasing muscle proportion of the bladder, higher smooth muscle actin (SMA) expression, and neuronal reinnervation. Moreover, Lee et al. investigated the therapeutic effect and possible mechanisms of LESW on diabetic bladder dysfunction in a rat model. They found that LESW eased bladder dysfunction and urinary continence, reflected by the restoration of the nerve expression of the urethra and the vascularization of the bladder [123]. However, a clinical report of LESW on patients with DU is still lacking.

### 10.7. Drug Delivery

Codama et al. reported that SWs produced a shear force, increased the tissue permeability, and facilitated transportation of pharmaceutical molecules into cells [28]. OnabotulinumtoxinA (onaBoNTA) intradetrusor injection is the third line treatment for OAB. Due to the large size of onaBoNTA (150 KDa), direct bladder instillation was shown to have no effect with an intact urothelium [125]. Chuang et al. used magnetic resonance imaging to demonstrated bladder urothelial leakage of Gd-DTPA after low energy SWs, which was not seen in controls [48]. Moreover, rats pretreated with botulinum toxin A plus low energy SWs showed a decreased inflammatory reaction and decreased expression of SNAP-23, SNAP-25, and COX-2, compared to the control group, in an acetic acid-induced bladder hyperactivity rat model.

A single arm prospective clinical study was conducted in 15 patients with refractory OAB receiving intravesical instillation of 100 IU of BoNT-A, followed by LESW (3000 shocks greater than 10 min) exposure to the suprapubic area. The results showed significant improvements in OABSS after 1 and 2 months, without compromising voiding function [126]. These results inspire clinical experience for following studies.

Luo et al. evaluated the synergistic effects of LESW and cisplatin in upper urinary tract urothelial carcinoma (UTUC). They reported that LESW increased tissue permeability and enhanced cisplatin cytotoxicity in UTUC cells, and suppressed tumor cell proliferation both in vitro and in vivo. Furthermore, the anti-tumor effect was enhanced in the human-derived organoid model, by interfering with ZO-1 and E-cadherin to increase the permeability of cisplatin into cancer cells [49]. Elkashef et al. also reported that LESW enhanced intravesical epirubicin penetration in a rat model of bladder cancer with decreasing p53, IL-6, miR-210, HIF-1α, and TNF-α expression, and better histopathological results [127]. Taken together, ESWT might be applied as a tool for drug delivery for bladder dysfunction and urothelial tumors. LESWs assisted drug delivery in uro-oncology is an interesting suggestion, but no comprehensive clinical trials have been conducted as of yet.

## 11. Conclusions

SW medicine has been gaining attention for urological applications in which shock induces various biological effects and might assist with restoration of body function. However, the mechanisms and molecular changes after LESW remain mostly unknown, and large clinical studies are still lacking. The applications of LESW on new frontiers of urology are promising, but more evidence is necessary before LESW can be widely used in daily practice.

## Figures and Tables

**Figure 1 biomedicines-10-00675-f001:**
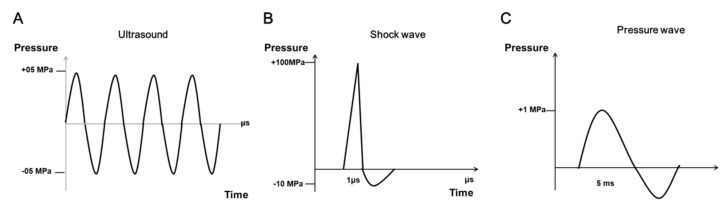
Typical types of ultrasound, focused shock wave, and pressure wave. (**A**) Ultrasound waves are periodic oscillations with a limited bandwidth. (**B**) Focused shock wave is characterized by a single positive pressure wave (100 MPa), followed by a negative pressure wave (−10 MPa). The 1 microsecond (μs) of time interval is the positive pressure wave. (**C**) The pressure wave is 1 MPa positive wave and less than 1 MPa negative wave in 5 milliseconds (ms).

**Figure 2 biomedicines-10-00675-f002:**
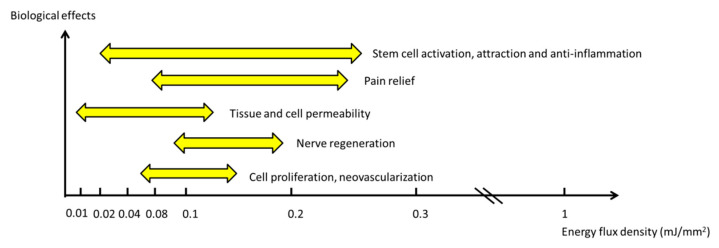
The biological effects associated with related energy levels of ESWT.

**Figure 3 biomedicines-10-00675-f003:**
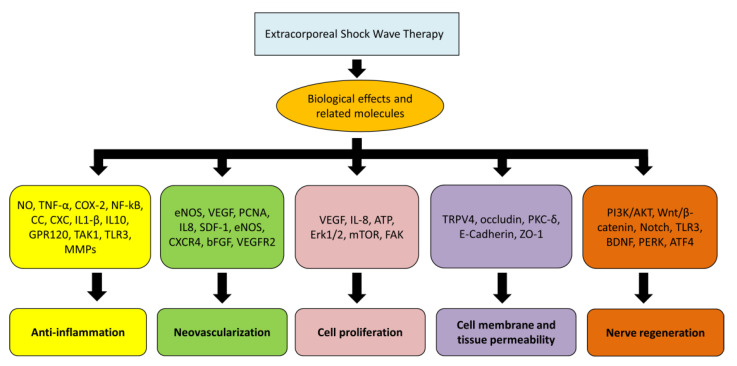
Biological effects and related molecules after ESWT.

**Table 1 biomedicines-10-00675-t001:** LESW for chronic prostatitis/chronic pelvic pain syndrome.

	Study Design	N	Treatment Setting Treatment Course (mJouls/mm^2^)	Following Duration (Weeks)	Treatment Effect
Zimmermann et al., 2008 [58]	Cohort	34	3000 impulses per week, 4 weeks. 0.25, 3 Hz.	1, 4, 12	Improvements in pain and QoL.
Zimmermann et al., 2009 [59]	Double-blind. RCT	60	3000 impulses per week, 4 weeks. 0.25, 3 Hz.	1,4,12	Improved QoL, IIEF, CPSI, VAS, and IPSS at 1, 4, and 12 weeks.
Zeng et al., 2012 [60]	RCT	80	3000 impulses, 5 times a week, 2 weeks. 0.06-maximum tolerated dose.	4, 12	Decreased NIH-CPSI score, improved QOL, 71.1% vs. 28.9% at week 2.
Vahdatpour et al., 2013 [61]	RCT	40	3000 impulses per week, 4 weeks. 0.25, 3 Hz, increased to 0.3, 0.35, 0.4.	1,2,3, 12	Pain scores decreases at 2, 3, and 12 weeks. Urinary scores improved in weeks 3 and 12.
Moayednia et al., 2014 [62]	RCT	19	3000 impulses per week, 4 weeks. 0.25, 3 Hz.	16, 20, 24	NIH-CPSI, VAS, IPSS decreased. Not statistically different at week 24.
Pajovic et al., 2016 [63]	RCT	30	3000 impulses per week, 4 weeks. 0.25, 3 Hz.	12, 24, 36	Improved NIH-CPSI. No change for PVR or QMAX.
Al Edwan et al., 2017 [64]	Cohort	41	2500 impulses per week, 4 weeks. 0.25, 3 Hz.	2, 24, 48	Improved NIH-CPSI, IIEF, IPSS.
Guu et al., 2018 [65]	Cohort	33	3000 impulses per week, 4 weeks. 0.25, 3 Hz.	1, 4, 12	Improved NIH-CPSI, IIEF, IPSS.
Salama et al., 2018 [66]	RCT	40	3000 pulses, 12 Hz at 3 to 5 bar, twice a week for 4 weeks.	1, 4, 8	Four domains of the NIH-CPSI decreased at weeks 1, 4, and 8.
Zhang et al., 2019 [67]	nRCT	40	3000 pulses per week, 8 weeks. 1.8–2.0 bar; 10 Hz.	4, 8, 12	Both groups improved NIH-CPSI, QoL, VAS, IPSS, and IIEF-5. Only 8 weeks CPSI. IIEF statistically significant.
Skaudickas et al., 2020 [68]	Cohort	40	3000 impulses per week, 4 weeks. 0.25, 3 Hz.	0, 4, 12	NIH-CPSI, IPSS, VAS, and IIEF-5 showed greatest improvement at week 4; VAS and IPSS, improvement at week 12.
Li et al., 2020 [69]	Cohort	32	3000 impulses per week, 4 weeks. 0.25, 3 Hz.	1, 2, 4, 12	VAS and the NIH-CPSI showed substantial improvement at week 4 and 12.
Kim et al., 2021 [70]	Double-blind RCT	34	3000 impulses per week, 8 weeks. 0.25, 3 Hz.	0, 4	NIH-CPSI, QoL, IIEF, and VAS decrease at Week 0 and 4.
Mykoniatis et al., 2021 [71]	Double-blind RCT	45	5000 shockwaves per week, 6 weeks, 0.1.	4, 12, 24	NIH-CPSI, pain, and QoL improved, persisted at 24 weeks. No improvement of NIH-CPSI urinary subdomain and IPSS.
Sakr et al., 2021 [72]	RCT	155	3000 impulses per week, 4 weeks. 0.25, 3 Hz.	4, 12, 24, 48	NIH-CPSI, IPSS, VAS, and IIEF-5.
Wu et al., 2021 [73]	Cohort	215	3000 impulses per week, 6 weeks. 0.25, 4 Hz.	4, 12, 24, 48	Improved NIH-CPSI, IIEF, IPSS, and AUA QoL_US at 4, 12, 24, and 48 weeks

CPSI = Chronic Prostatitis Symptom Index; IIEF = International Index of Erectile Function; IPSS = International Prostate Symptom Score; VAS = Visual Analog Scale; AUA QoL_US = American Urological Association Quality of Life due to Urinary Symptom.

**Table 2 biomedicines-10-00675-t002:** LESW for erectile dysfunction.

	Study Design	N	Treatment Setting Treatment Course (mJouls/mm^2^)	Following Duration (Months)	Treatment Effect
Skolarikos et al., 2005 [90]	Cohort	40	3000 impulses, 6 weeks.	3, 12	64.2% improve IIEF Improvement in penile angulation, with a mean reduction of 35 degrees No significant change in plaque size
Poulakis et al., 2006 [91]	RCT	68	2000 impulses per week, 5 weeks, 0.25.	1, 3, 6	Improvement in pain, IIEF-5 score and plaque size, but no difference compared to another group.
Zimmermann et al., 2009 [59]	RCT	60	3000 impulses per week, 4 weeks, 0.25, 3 Hz.	1, 3	Improvement of pain, QoL, and voiding conditions IIEF-5.
Chitale et al., 2010 [92]	RCT	36	3000 impulses per week, 6 weeks.	3, 6	Improved IIEF-5 and VAS score. No significant change in Peyronie’s disease.
Gruenwald et al., 2012 [93]	Cohort	29	1500 impulses twice per week, 3 weeks, 0.09.	1, 2	DE-5i poor responders. Improved EHS and IIEF-ED.
Vardi et al., 2012 [94]	RCT	67	1500 impulses twice per week, 9 weeks, 0.09.	1	Improved IIEF-5, EHS, and penile blood flow.
Palmieri et al., 2012 [95]	Cohort	50	2000 impulses per week, 4 weeks, 0.25.	3, 6	Improved IIEF-5 and quality of life.
Yee et al., 2014 [96]	RCT	70	1500 impulses twice per week, 9 weeks, 0.09.	1	Clinical improvement in IIEF-ED and EHS, but no significant difference between two groups.
Srini et al., 2015 [97]	RCT	135	NA	1, 3, 6, 9, 12	Improvement in IIEF-EF, EHS, and CGIC.
Pelayo-Nieto et al., 2015 [98]	Cohort	15	5000 impulses per week, 4 weeks, 0.09.	1, 6	Improvement in IIEF, SEP, and GAQ.
Chung and Cartmill 2015 [99]	Cohort	30	3000 impulses twice per week, 6 weeks, 0.25.	1, 4	PDE5i non-responders; 18 (60%) patients ≥ 5 points improvement in IIEF-5 score; 21 (70%) patients >50% in EDITS index score, last 4-months.
Bechara et al., 2015 [100]	Cohort	25	5000 impulses once per week, 4 weeks, 0.09.	3	PDE-5i non-responders. Improved IIEF-6, SEP2, SEP3, and GAQ. Restoring PDE5i response in >50% of patients.
Frey et al., 2015 [101]	Cohort	18	3000 impulses twice per week, 6 weeks. 20, 15, and 12.	1, 12	post-prostatectomy erectile dysfunction. Improved IIEF-5 scores at 1 and 12 months.
Olsen et al., 2015 [102]	RCT	112	3000 impulses per week, 5 weeks, 0.15.	1, 3, 6	Improved IIEF-5 and EHS.
Hisasue 2016 [103]	Cohort	57	1500 impulses twice per week, 9 weeks, 0.09.	1, 3, 6	Improvement in IIEF, EHS, and MPCC.
Kalyvianakis et al., 2018 [104]	RCT	44	5000 impulses once/twice per week, 6 weeks, 2 phase treatment, 0.05.	1,3, 6	Vasculogenic ED, PDE5 responders. Improvement in IIEF-EF score, MCID, and SEP3 score. 12 sessions twice per week were better than 6 sessions once a week.
Vinay et al., 2021 [105]	RCT	76	5000 impulses once per week, 4 weeks, 0.09.	1, 3, 6	*PDE5I refractory patients. Improved IIEF-EF in 3 and 6 months.
Oliveira et al., 2019 [106]	Cohort	25	2000 impulses on perineum + 2000 on dorsum penis once per week, 6 weeks, 0.16.	1.5, 3	Improved PSV and EDV. Disease duration dose not negative impact on treatment outcomes.
Huang et al., 2020 [107]	Cohort	35	NA	1,	Improved IIEF-5, EHS, erectile rigidity, and nocturnal erection frequency.
Sramkova et al., 2020 [108]	RCT	60	6000 impulses twice per week, 2 weeks, 0.160.	1, 3	Improve IIEF-5, EHS, GAQ, SEP 2, and SEP 3.
Palmieri et al., 2021 [109]	RCT	106	3000 impulses twice per week, 3 weeks, 0.25.	1	vasculogenic ED PDE5i non-responders. Improved IIEF-EF, PSV, and EDV.
Shendy et al., 2021 [110]	RCT	42		3	Improved IIEF-5 and PSV.

GAQ = Global Assessment Questionnaire EHS = Erectile Hardness Score, CGIC = Clinical Global Impression of Change, MPCC = maximal penile circumferential change, NA = not available, SEP3 = Sexual Encounter Profile question 3 score, MCIDs = minimally clinical important differences.

## Data Availability

Not applicable.

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
