# Peer review of "New Frontiers of Extracorporeal Shock Wave Medicine in Urology from Bench to Clinical Studies"

_biomedicines, 2022, doi:10.3390/biomedicines10030675_

Round 1
Reviewer 1 Report
Aim of this review is to discuss the role and potential future developments of shock waves in Urology. The aim is clear and interesting due to the increasing applications of ESWT in medicine and urology. The manuscript is well written and tables and figures are informative.
Author Response
Reviewer 1
Comments and Suggestions for Authors
Aim of this review is to discuss the role and potential future developments of shock waves in Urology. The aim is clear and interesting due to the increasing applications of ESWT in medicine and urology. The manuscript is well written and tables and figures are informative.
A: We appreciated the reviewer’s positive comments.

Reviewer 2 Report
I would like to suggest that new frontiers implicates already defined issues, which this is most certainly not. Most of the studies included comprehending small groups and above all, animal models for support. Most of the studies have the conclusion as "should be improved" of "potential for clinical application". Some clinical studies are not finished, some of them require further follow-up. The title should include a review of animal and clinical studies. Therefore abstract should be revised according to the conclusions section which is quite sincere. For example "LESWs assisted drug delivery" is an interesting suggestion, but no comprehensive clinical trials have been conducted as of yet. Every subsection must also contain the limitations of each study. The conclusion is well defined and should be reflected to the whole paper.Author Response
Reviewer 2
Comments and Suggestions for Authors
- I would like to suggest that new frontiers implicate already defined issues, which this is most certainly not. Most of the studies included comprehending small groups and above all, animal models for support. Most of the studies have the conclusion as "should be improved" of "potential for clinical application". Some clinical studies are not finished, some of them require further follow-up. The title should include a review of animal and clinical studies.
A: We appreciated the reviewer’s comments and revised the title as “New Frontiers of Extracorporeal Shock Wave Medicine in Urology- Implications from Bench to Clinical Studies”.
- Therefore abstract should be revised according to the conclusions section which is quite sincere. For example "LESWs assisted drug delivery" is an interesting suggestion, but no comprehensive clinical trials have been conducted as of yet.
A: We appreciated the reviewer’s comments and revised the abstract as “LESWs assisted drug delivery in uro-oncology is an interesting suggestion, but no comprehensive clinical trials have been conducted as yet. Taken together, LESW is a promising method for the treatment of various diseases in urology. However, further investigation with large scale of clinical studies is necessary to confirm the real role LESW in clinical use. This article provides information on the basics of SW physics, mechanisms of action on biological systems, and new frontiers of SW medicine in urology.”
- Every subsection must also contain the limitations of each study. The conclusion is well defined and should be reflected to the whole paper.
A: We appreciated the reviewer’s comments and add limitations at the end of every subsection if not provided.
10.1 Although these studies provided clinical evidence for the efficacy of ESWT, the mechanisms of its action from human prostate specimen and biomarkers remained to be determined.
10.2 However, the beneficial effects of ESWT on pre-clinical studies of radiation cystitis and ketamine cystitis warrant that larger multi-institutional studies with placebo control are needed.
10.5 However, larger scale of clinical study is necessary to elucidate the real role of LESW for the treatment of SUI.
10.7 LESWs assisted drug delivery in uro-oncology is an interesting suggestion, but no comprehensive clinical trials have been conducted as yet.
Round 2
Reviewer 2 Report
My previous review was not meant to be taken literally, but as a sentence suggestion. Nevertheless, I would change in the abstract and drug delivers sections the phrase "an interesting suggestion" into "promising". Otherwise, I have no further comments.